# Physical activity and diet pattern do not mitigate C-reactive protein increases associated with oral contraceptive use

Eric T. Trexler[1]*, David E. Eagle[2,3], Herman Pontzer[1,2]

1 Department of Evolutionary Anthropology, Duke University, Durham, North Carolina, United States of America, 2 Duke Global Health Institute, Duke University, Durham, North Carolina, United States of America, 3 Department of Sociology, Duke University, Durham, North Carolina, United States of America

* eric.trexler@duke.edu

## Abstract

The purpose of this study was to examine the influence of body mass index (BMI), physical activity (PA) level, dietary inflammatory index (DII), and oral contractive (OC) use on C-reactive protein (CRP) levels, and to determine if elevated CRP values reflect systemic inflammation in OC users. Data were obtained from four cycles (1999-2006) of the U.S. National Health and Nutrition Examination Survey (NHANES) study, yielding a sample of 496 current OC users and a comparator group of 1,583 regularly menstruating women. A general linear model was used to test for interaction effects among BMI, PA level, and OC use, after adjusting for age and smoking status, with log-transformed CRP (lnCRP) identified as the outcome variable. Sequential general linear models with no interaction terms were then constructed to examine the impact of BMI, PA level, and OC use on circulating lnCRP after adjusting for age and smoking status. Follow-up analyses used general linear models to assess the relationship between lnCRP and other indices of systemic inflammation among OC users and nonusers, and to examine the predictors of lnCRP within each subgroup. The omnibus model including smoking status, age, PA level, OC use, and BMI did not identify any statistically significant two-way or three-way interaction effects (all $p \geq .259$). The adjusted $r^2$ value of the model modestly increased from 0.3789 to 0.3801 when all interaction terms were removed. After adjusting for smoking status and age, a sequentially built model indicated that PA level was inversely related to lnCRP values ($p = .0019$). When OC use was added to the model, it was positively associated with lnCRP values ($p < .0001$), with statistically and clinically significant lnCRP differences between OC users and nonusers. BMI was the last variable entered into the model, which was positively associated with lnCRP ($p < .0001$). Among OC nonusers, PA level ($p = .0008$) and BMI ($p < .0001$) were significantly predictive of lnCRP levels after adjusting for smoking status and age. In contrast, PA level was not significantly predictive of lnCRP values ($p = .718$) among OC users. All alternative indices of inflammation were positively correlated with lnCRP values (all $p < .0001$), but correlations were significantly stronger among OC users than nonusers (all $p < .05$). In a subset of OC nonusers with complete nutrition data, PA level ($p = .021$), BMI ($p < .0001$), and DII ($p = .007$) were significantly predictive of lnCRP after adjusting for smoking status and age. In contrast, PA level ($p = .709$)

**Data availability statement:** All data included in this study are publicly available on the US National Center for Health Statistics webpage (https://www.cdc.gov/nchs/nhanes/index.htm). Analytical code for this study is publicly accessible via the Open Science Framework website (https://osf.io/exv3q/).

**Funding:** The author(s) received no specific funding for this work.

**Competing interests:** The authors have declared that no competing interests exist.

and DII ($p = .690$) were not significantly predictive of lnCRP values among OC users. In conclusion, OC-induced elevations in CRP appear to be reflective of a chronic, systemic inflammatory response. PA and low DII are associated with lower CRP among OC nonusers, but do not mitigate CRP elevations among OC users.

## Introduction

Chronic inflammation plays a causative role in the development and progression of many diseases, including cardiovascular disease, cancer, diabetes, kidney disease, non-alcoholic fatty liver disease, autoimmune diseases, and several neurodegenerative disorders [1]. Medical conditions associated with chronic inflammation pose a significant public health concern, collectively constituting the largest drivers of global morbidity and mortality [1] and placing a substantial burden on healthcare systems [2]. Monitoring inflammation biomarkers and designing interventions to mitigate chronic inflammation are therefore important for improving health outcomes at the individual and population level. C-reactive protein (CRP) is an acute-phase protein released from the liver in response to inflammatory insults. It is one of the most sensitive and commonly researched biomarkers of systemic inflammation [3], and several studies have indicated that CRP levels are positively associated with inflammation-related chronic diseases [4]. Circulating CRP values below 1.0 mg/L represent low cardiovascular risk, whereas values between 1.0 to < 3.0 mg/L indicate moderate risk, values of 3.0 or higher indicate high risk, and values above 10.0 indicate very high risk [5].

Approximately 151 million women use oral contraceptives (OCs) according to recent estimates [6]. The most common formulations include a combination of exogenous estrogen and progestin, which effectively prevent pregnancy by providing to negative feedback to the hypothalamic-pituitary-gonadal axis [7]. By inhibiting the release of gonadotropin-releasing hormone, follicle-stimulating hormone, and luteinizing hormone, exogenous estrogen and progestin inhibit follicular development and prevent ovulation while reducing menstrual bleeding. A wide range of OC formulations have been approved for use in the United States using varying forms and dosages of exogenous estrogen and progestin, including progestin-only formulations. OCs are prescribed widely and are generally deemed to have an acceptable safety profile [7], but previous studies have documented statistically and clinically significant elevations of CRP among young, healthy OC users [8–10]. A crossover trial by Johnson et al. indicated that the relationship between OCs and CRP is causal in nature, with CRP levels approximately quadrupling after only two months of OC use [11]. While physical activity (PA) [12], low adiposity [13], and anti-inflammatory diet patterns [14] have been associated with CRP reductions, emerging studies suggest that OC-induced CRP elevations persist across a range of PA and adiposity levels [13]. Even among world-class endurance and team sport athletes, CRP levels are significantly elevated in OC users [15]. It is possible that acute stressors of intense training could contribute to these observed CRP elevations, but direct comparisons to OC nonusers engaged in similar training cast doubt on this potential explanation [15]. Preliminary studies appear to suggest that OC users are unable to fully mitigate elevations in CRP by maintaining low adiposity and high PA levels, but these observations are based on very few studies with relatively small sample sizes and homogenous populations, such as university students or athletic teams.

Given the well-established link between CRP and systemic inflammation, the persistence of OC-induced CRP elevations in lean and highly active athletes may suggest that PA and low adiposity are insufficient for mitigating systemic inflammation induced by OCs. Alternatively, these observations of high CRP levels in athletes may suggest that OC-induced CRP elevations

are not reflective of systemic inflammation. In support of this concept, a randomized cross-over trial by Van Rooijen et al. [16] previously reported significant CRP elevations without concomitant increases in interleukin-6 or tumor necrosis factor α in response to a combined OC containing ethinyl estradiol, which led the researchers to conclude that the observed CRP elevations were not indicative of a systemic inflammatory response. There are several mechanisms by which OCs could increase CRP [17], which may include systemic inflammation or localized effects on hepatic production of CRP. Elevations in CRP are primarily driven by exogenous estrogen rather than progestin [18,19], and previous studies have documented that vaginal [20] and transdermal [21] routes of administration attenuate the CRP elevations observed with oral administration of exogenous estrogen. These observations may suggest that first-pass metabolism of orally ingested estrogen increases hepatic CRP synthesis in the absence of systemic inflammation, but more research is needed to resolve discrepancies in the existing literature. For example, some studies with contradictory findings have reported that OC-induced CRP elevations are correlated with other biomarkers associated with a systemic inflammatory response, such as blood hydroperoxides [22] and interleukin-6 [20], and that oral and transdermal hormonal contraceptives yield similar CRP elevations [11].

There are currently several unresolved questions pertaining to the clinical implications of CRP elevations induced by OC use. Most notably, it is unclear if OC-induced elevations in CRP reflect a systemic inflammatory response and if these elevations can be mitigated or attenuated by PA, low body mass index (BMI), or anti-inflammatory diet patterns in the general population. The primary aim of this study was to examine associations among PA level, OC use, BMI, and circulating CRP levels after adjusting for age and smoking status. The secondary aim was to assess the relationship between CRP and other indices of systemic inflammation among OC user and nonuser subgroups, and the tertiary aim was to determine if BMI, PA level, and dietary inflammatory index (DII) are predictive of CRP levels among OC user and nonuser subgroups. We hypothesized that 1) OC use and BMI would be positively associated with CRP levels in the full sample; 2) correlations between CRP and other indices of inflammation would be decoupled among OC users (reflecting OC-induced CRP elevations in the absence of a systemic inflammatory response); 3) PA level would only be associated with CRP levels among OC nonusers; and 4) DII would only be associated with CRP levels among OC nonusers. These hypotheses collectively reflect the perspective that OC-induced elevations in CRP are not indicative of systemic inflammation, and are therefore unresponsive to the anti-inflammatory effects of PA and low DII.

## Materials and methods

### Participants

Data for the current study were obtained through the U.S. National Health and Nutrition Examination Survey (NHANES). The U.S. Centers for Disease Control and Prevention (CDC) has conducted continuous sampling for this survey since 1999 and makes the resulting data publicly available in two-year cycles [23]. Data collection methods were approved by the CDC/NCHS Ethics Review Board, and all study participants provided written informed consent. For the present analysis, data were obtained from four NHANES cycles (1999–2000, 2001–2002, 2003–2004, and 2005–2006), which include a total of 41,474 participants. Cycles beyond 2006 were not included due to changes in the PA questionnaire in 2007, followed by a lack of CRP data from 2011–2014 and changes to the reproductive health questionnaire in 2013. The sample was restricted to female participants between the ages of 18 and 65 who were either currently using OCs or regularly menstruating. The sample was further restricted to individuals with complete data available for key variables of interest (CRP, age, smoking

status, PA level, OC use, BMI, platelet count, neutrophil count, monocyte count, and lymphocyte count).

## Measures

### Physical activity level

Survey item PAQ180 was used to quantify PA level. This item asks participants to qualitatively select an activity category (ranging from 1–4) that corresponds with their typical PA level, with higher values reflecting increasing PA levels. Previous research has validated this questionnaire against accelerometry counts [12]. In line with previous work in this area [12], PA level was recoded to combine categories 3 and 4, as only 61 participants (<3% of the sample) selected category 4.

### C-reactive protein

C-reactive protein values were obtained via latex-enhanced nephelometry. Values were converted from mg/dL to mg/L to facilitate interpretation and comparison against commonly used cardiovascular risk thresholds (low risk, < 1.0 mg/L; moderate risk, 1.0 to < 3.0 mg/L; high risk, ≥ 3.0 mg/L; very high risk, > 10.0 mg/L). Due to substantial skewness, CRP values were transformed for all analyses by calculating the natural logarithm (lnCRP).

### Contraceptive use and menstrual status

Information pertaining to OC use and menstrual status was obtained from the reproductive health questionnaire. In 1999–2000 and 2001–2002 cycles, OC use was assessed by item RHD440 ("Are you taking birth control pills now?") and menstrual status was assessed by item RHQ030 ("Have you had regular periods in the last 12 months?"). In 2003-2004 and 2005-2006 cycles, OC use was assessed by item RHD442 ("Are you taking birth control pills now?"). Survey questions about OC use specifically refer to current use, so duration of OC use is unknown for survey respondents. These NHANES questionnaires also provide incomplete information regarding the exact type, formulation, and generation of OC used by each study participant. As a result, the present analysis is not able to distinguish between progestin-only versus combined formulations, monophasic versus multiphasic formulations, different forms or generations of exogenous estrogen and progestin, or different dosages of exogenous estrogen and progestin. Reproductive questionnaires used in the 1999-2006 NHANES cycles included item RHQ540, which asked participants if they have ever used female hormone therapy for reasons other than contraception or infertility treatment. As such, it is highly unlikely that a meaningful number of participants undergoing hormone replacement therapy would mistakenly self-identify as using birth control pills.

Menstrual status was assessed by a survey item that asked participants, "When did you have your last period?" This survey item was identified as RHQ050 from 1999–2002 and as RHQ051 from 2003–2006. For the present analysis, participants were considered "regularly menstruating" if they selected "having it now" or "less than 2 months ago" for this survey item, as answers beyond 2 months are more likely to reflect clinically relevant menstrual cycle disorders. Survey items related to OC use and menstrual status do not identify the current menstrual cycle phase or distinguish between the active or inactive phase of OC pills at the time of each participant's NHANES assessment. However, this is unlikely to introduce systematic bias to our analysis, and previous literature indicates that CRP fluctuations across the menstrual cycle are small in magnitude and that OC-induced CRP elevations persist across all OC pill phases [24].

## Alternative indices of inflammation

Systemic immune-inflammation index (SII), systemic inflammation response index (SIRI), and systemic immune-inflammation response index (SIIRI) are previously validated indices of systemic inflammation [25]. All three indices were calculated to examine the relationship between CRP values and alternative inflammation metrics. Platelet, neutrophil, monocyte, and lymphocyte data were obtained from NHANES complete blood count data to calculate inflammation indices as shown in equations (1), (2), and (3).

$$SII = \frac{Platelet\ count\ \times\ Neutrophil\ Count}{Lymphocyte\ Count} \tag{1}$$

$$SIRI = \frac{Monocyte\ count\ \times\ Neutrophil\ Count}{Lymphocyte\ Count} \tag{2}$$

$$SIIRI = \frac{Platelet\ count\ \times\ Monocyte\ count\ \times\ Neutrophil\ Count}{Lymphocyte\ Count} \tag{3}$$

## Dietary inflammatory index

The dietary inflammatory index (DII) is a validated index designed to quantify the cumulative anti-inflammatory or pro-inflammatory impact of a given diet, with negative values representing anti-inflammatory effects and positive values representing pro-inflammatory effects [26]. R package "dietaryindex" is a validated informatics tool for standardized dietary index calculations [27]. For NHANES 24-hour diet recall data beginning in the 2005–2006 cycle, this tool uses the Food Patterns Equivalents Database to calculate scores for a variety of diet indices and scoring systems. The Food Patterns Equivalents Database was created in 2005, so the present dietary analysis was restricted to data from the 2005–2006 NHANES cycle (n = 578). While NHANES 24-hour dietary recalls provide most of the information used in the calculation of DII, the "dietaryindex" tool calculates NHANES DII values without the inclusion of vitamin D, eugenol, garlic, ginger, onion, trans fat, turmeric, green/black tea, flavan-3-ol, flavones, flavonols, flavonones, anthocyanidins, isoflavones, pepper, thyme/oregano, and rosemary. For the present analysis, DII was calculated using data from both the first and second 24-hour dietary recalls.

## Anthropometric data, demographic characteristics, and smoking status

The impact of adiposity was modeled using body mass index (BMI), which was obtained directly from the NHANES examination data. Previous research has indicated that age and smoking status impact CRP levels, so these variables were collected for inclusion as covariates. Age was obtained from NHANES demographic data. Smoking status was coded by combining items SMQ020 ("Have you smoked at least 100 cigarettes in your entire life?") and SMQ040 ("Do you now smoke cigarettes?"). Participants were considered current smokers if they reported currently smoking "every day" or "some days."

## Statistical analysis

### Hypothesis 1

Bivariate relationships between predictors of interest (PA level, OC use, and BMI) and the outcome variable (lnCRP) were assessed using unadjusted linear models. To test for interaction

effects, the omnibus multivariable model included PA level, OC use, BMI, and all two-way and three-way interactions among them, adjusted for smoking status and age as covariates. All continuous variables were mean-centered, and *F* statistics for the interaction model were calculated using partial (type III) sums of squares. In the absence of statistically significant interaction effects, simplified models without interaction terms were constructed. To attenuate the statistical impact of correlated predictor variables, the simplified models were constructed in a sequential manner, with *F* statistics calculated using sequential (type I) sums of squares. The order of variable entry was determined *a priori* and guided by the primary research question and corresponding hypothesis; covariates (smoking status and age) were entered into the model first, followed by PA level, OC use, and BMI.

### Hypothesis 2

A series of linear models were constructed to examine bivariate relationships between lnCRP and alternative indices of inflammation (SII, SIRI, and SIIRI). For each bivariate relationship, Fisher's *z* transformation was used to statistically test whether there was a significant difference in the strength of the relationship among OC users in comparison to the strength of the bivariate relationship observed among OC nonusers.

### Hypothesis 3

The sample was split into subgroups (OC users and nonusers) to facilitate preplanned analyses addressing predictors of lnCRP values within each subgroup. These models were constructed in a sequential manner, with *F* statistics calculated using sequential (type I) sums of squares. The order of variable entry was determined *a priori* and guided by the secondary research questions and corresponding hypotheses; covariates (smoking status and age) were entered into the model first, followed by PA level and BMI.

### Hypothesis 4

Sequential linear models were used to examine the association between DII and ln-transformed CRP levels within each subgroup (OC users and nonusers). The order of variable entry was determined *a priori* and guided by the secondary research questions and corresponding hypotheses; covariates (smoking status and age) were entered into the model first, followed by PA level, BMI, and DII. *F* statistics were calculated using sequential (type I) sums of squares. All analyses were planned *a priori* with a significance threshold of $\alpha = .05$ and conducted using R software (version 4.2.1).

## Results

After applying inclusion and exclusion criteria, the sample included 2,079 participants.

### Hypothesis 1

Unadjusted bivariate analyses indicated that PA level ($F$[2, 2076] = 6.11, $p = .0023$), OC use ($F$[1, 2077] = 110.70, $p < .0001$), and BMI ($F$[1, 2077] = 781.27, $p < .0001$) were significantly predictive of lnCRP levels. Log-transformed CRP values were negatively associated with PA levels (estimated marginal mean ± standard error; low, 0.872 ± 0.06 mg/L; medium, 0.783 ± 0.04 mg/L; high, 0.566 ± 0.07 mg/L). Estimated geometric means for each PA level, back-transformed to the original scale, are presented in Fig 1A. Log-transformed CRP values were higher among OC users (1.31 ± 0.06 mg/L) than nonusers (0.59 ± 0.03 mg/L). Estimated geometric means for OC users and nonusers are presented in Fig 1C. Each 1-unit increase in BMI

was associated with an increase in log-transformed CRP values of 0.097 ± 0.003 mg/L (Fig 1E). The exponentiated value of this regression coefficient indicates that a 1-unit increase in BMI is associated with a 10.1% increase in CRP.

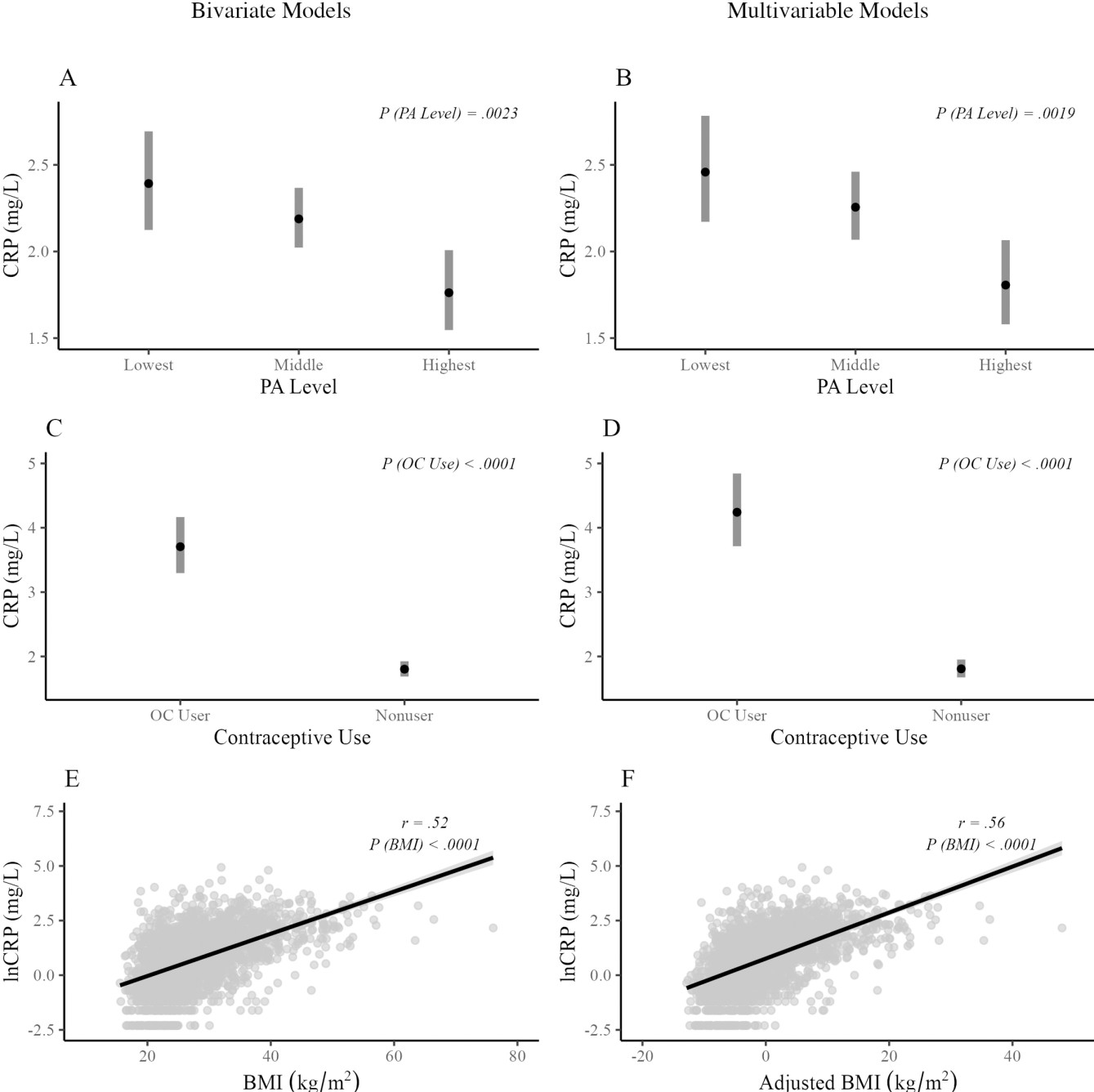

**Fig 1. Effects of PA level, OC use, and BMI on predicted C-reactive protein values in bivariate (A, C, E) and multivariable (B, D, F) models.** Panels A, B, C, and D present geometric mean estimates and 95% confidence intervals for C-reactive protein transformed back to the original scale (mg/L). Panel D presents the relationship between OC use and CRP after asking for smoking status, age, and PA level. Panel F visualizes the relationship between BMI and lnCRP after adjusting for smoking status, age, PA level, and OC use. CRP, C-reactive protein; PA, physical activity; OC, oral contraceptive; BMI, body mass index; lnCRP, natural logarithm of C-reactive protein.

The omnibus model including smoking status, age, PA level, OC use, and BMI did not identify any statistically significant two-way or three-way interaction effects (all $p \geq .259$). The adjusted $r^2$ value of the model modestly increased from 0.3789 to 0.3801 when all interaction terms were removed. After adjusting for smoking status and age, a multivariable model with no interaction terms indicated that PA level had a statistically significant

**Table 1. Sample descriptive statistics.**

|  | Full sample (n = 2079) | OC users (n = 496) | OC nonusers (n = 1583) |
|---|---|---|---|
| **Age, yr** | 35 ± 9 | 30 ± 8 | 37 ± 8 |
| **BMI, kg . m$^{-2}$** | 28.3 ± 7.4 | 26.2 ± 6.2 | 28.9 ± 7.6 |
| **CRP, mg/L** | 2.4 (0.8, 5.9) | 4.2 (1.7, 8.5) | 2.0 (0.7, 4.9) |
| **CRP risk category** |  |  |  |
| Low (>1.0 mg/L) | 608 (29.2%) | 79 (15.9%) | 529 (33.4%) |
| Moderate (1.0 to < 3.0 mg/L) | 563 (27.1%) | 119 (24.0%) | 444 (28.0%) |
| High (≥3.0 mg/L) | 908 (43.7%) | 298 (60.1%) | 610 (38.5%) |
| **Smoking status** |  |  |  |
| Smoker | 501 (24.1%) | 82 (16.5%) | 419 (26.5%) |
| Nonsmoker | 1578 (75.9%) | 414 (83.5%) | 1164 (73.5%) |
| **Physical Activity Level** |  |  |  |
| Lowest | 507 (24.4%) | 120 (24.2%) | 387 (24.4%) |
| Middle | 1153 (55.5%) | 283 (57.1%) | 870 (55.0%) |
| Highest | 419 (20.2%) | 93 (18.8%) | 326 (20.6%) |

Yr, years; BMI, body mass index; CRP, C-reactive protein; n, number of participants. Age and BMI presented as mean ± SD; CRP presented as median (interquartile range); CRP cardiovascular risk category, smoking status, and physical activity level presented as n (%). Descriptive statistics of the sample are presented in Table 1.

**Table 2. Model estimates of CRP by PA level, BMI, and OC use.**

|  | BMI = 20 | BMI = 25 | BMI = 30 | BMI = 35 |
|---|---|---|---|---|
| Lowest PA level |  |  |  |  |
| OC user | 1.97 (1.72, 2.25) | 3.33 (2.93, 3.79) | 5.63 (4.95, 6.41) | 9.53 (8.31, 10.93) |
| Nonuser | 0.65 (0.58, 0.74) | 1.10 (0.99, 1.23) | 1.87 (1.68, 2.07) | 3.16 (2.84, 3.52) |
| Middle PA level |  |  |  |  |
| OC user | 1.99 (1.77, 2.23) | 3.36 (3.02, 3.75) | 5.69 (5.10, 6.35) | 9.63 (8.54, 10.86) |
| Nonuser | 0.66 (0.60, 0.72) | 1.11 (1.03, 1.21) | 1.89 (1.75, 2.03) | 3.19 (2.93, 3.47) |
| Highest PA level |  |  |  |  |
| OC user | 1.76 (1.53, 2.03) | 2.99 (2.60, 3.43) | 5.05 (4.39, 5.81) | 8.55 (7.36, 9.93) |
| Nonuser | 0.58 (0.52, 0.66) | 0.99 (0.88, 1.11) | 1.67 (1.50, 1.87) | 2.83 (2.51, 3.19) |

Data presented as estimate (95% confidence interval). CRP estimates correspond to a 35 year-old nonsmoker and are transformed back to the original scale (mg/L). CRP, C-reactive protein; PA, physical activity; OC, oral contraceptive; BMI, body mass index.

effect on lnCRP values ($F[2, 2074] = 6.26$, $p = .0019$). Estimated geometric means for each PA level, back-transformed to the original scale, are presented in Fig 1B. When OC use was added to the model, it was significantly predictive of lnCRP values ($F[1, 2073] = 139.45$, $p < .0001$). Estimated geometric means for OC users and nonusers are presented in Fig 1D. When BMI was added to the model, it was significantly predictive of lnCRP values ($F[1, 2072] = 1045.23$, $p < .0001$). Each 1-unit increase in BMI was associated with an increase in log-transformed CRP values of $0.105 \pm 0.003$ mg/L (Fig 1F). The exponentiated value of this regression coefficient indicates that a 1-unit increase in BMI is associated with an 11.1% increase in CRP. To facilitate interpretation, CRP estimates from the full model are presented in Table 2.

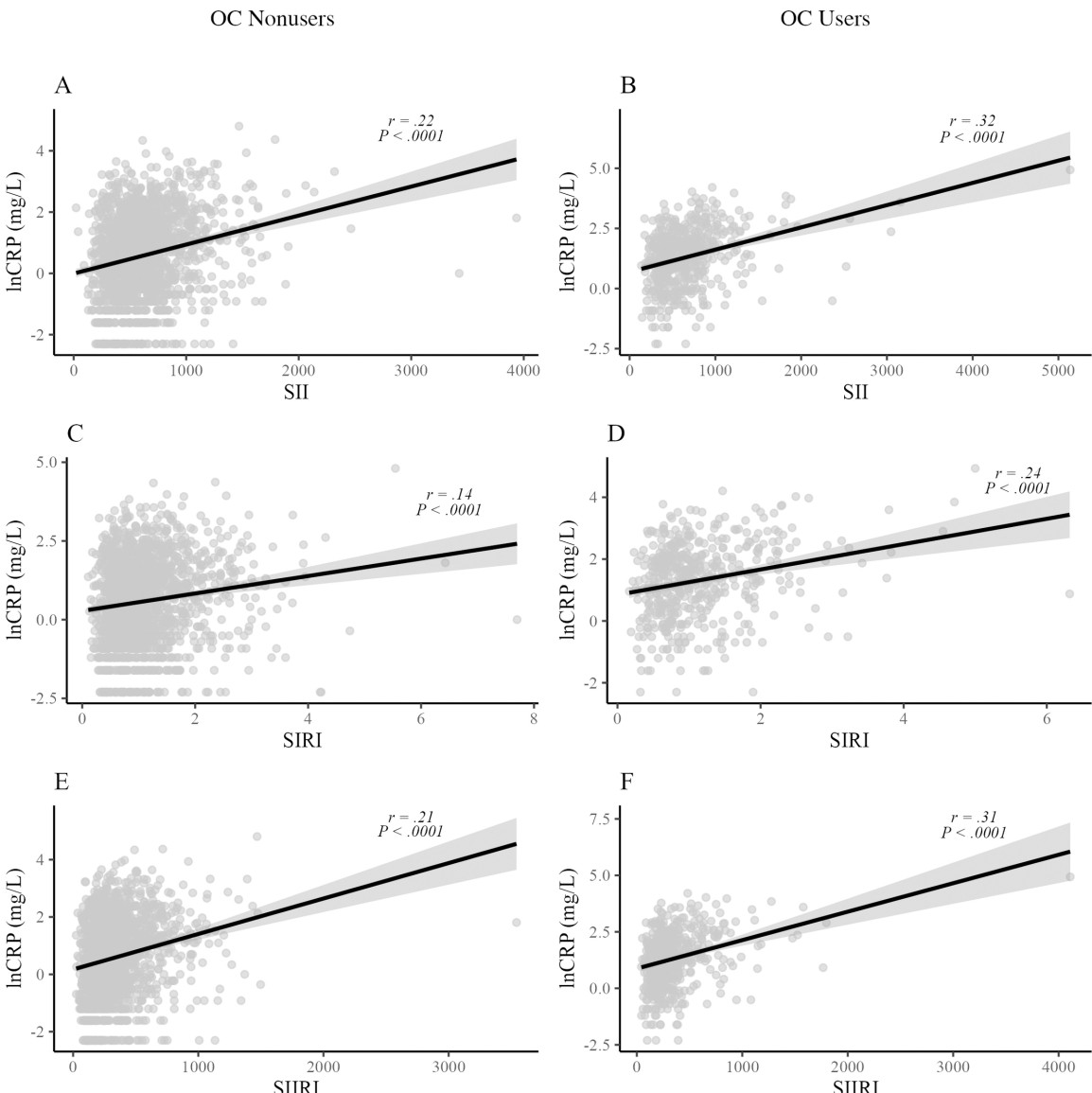

**Fig 2. Relationship between lnCRP and alternative indices of systemic inflammation among OC nonusers (A, C, E) and OC users (B, D, F). lnCRP, natural logarithm of C-reactive protein; SII, systemic immune-inflammation index; SIRI, systemic inflammation response index; SIIRI, systemic immune-inflammation response index.**

## Hypothesis 2

Among OC nonusers, lnCRP values were significantly correlated with all systemic inflammation indices: SII ($r = .22$, $p < .0001$), SIRI ($r = .14$, $p < .0001$), and SIIRI ($r = .21$, $p < .0001$). Log-transformed CRP values were significantly correlated with SII ($r = .32$, $p < .0001$), SIRI ($r = .24$, $p < .0001$), and SIIRI ($r = .31$, $p < .0001$) values among OC users. Relationships between lnCRP and alternative indices of systemic inflammation are presented in Fig 2. For SII ($z = -2.04$, $p = .041$), SIRI ($z = -2.15$, $p = .031$), and SIIRI ($z = -2.05$, $p = .040$), the correlation with lnCRP values was significantly stronger among OC users than nonusers.

## Hypothesis 3

After adjusting for smoking status and age, PA level was significantly predictive of lnCRP values ($F[2, 1578] = 7.16$, $p = .0008$) among OC nonusers. Estimated geometric means for each PA level, back-transformed to the original scale, are presented in Fig 3A. When BMI was added to the model, BMI was also significantly predictive of lnCRP values ($F[1, 1577] = 839.15$, $p < .0001$). A plot visualizing the independent relationship between BMI and lnCRP after adjusting for smoking status, age, and PA level is presented in Fig 3C.

Among OC users, PA level was not significantly predictive of lnCRP values ($F[2, 491] = 0.33$, $p = .718$). Estimated geometric means for each PA level, back-transformed to the original scale, are presented in Fig 3B. When BMI was added to the model, it was significantly predictive of lnCRP

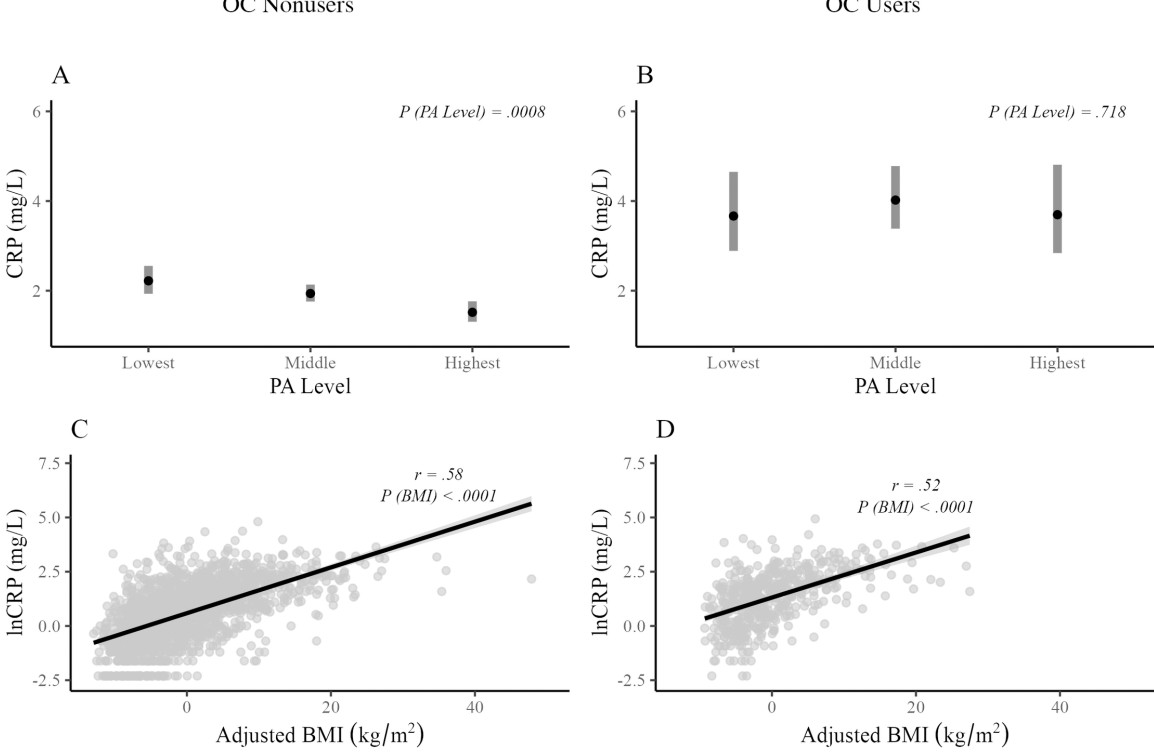

**Fig 3. Effects of PA level on predicted CRP values among OC nonusers (A) and users (C), and effects of BMI on predicted CRP values among OC nonusers (B) and users (D).** Panels A and B present geometric mean estimates and 95% confidence intervals for C-reactive protein transformed back to the original scale (mg/L). Plots in panels C and D visualize the independent relationships between BMI and lnCRP after adjusting for smoking status, age, and PA level among OC nonusers (C) and OC users (D). CRP, C-reactive protein; PA, physical activity; OC, oral contraceptive; BMI, body mass index; lnCRP, natural logarithm of C-reactive protein.

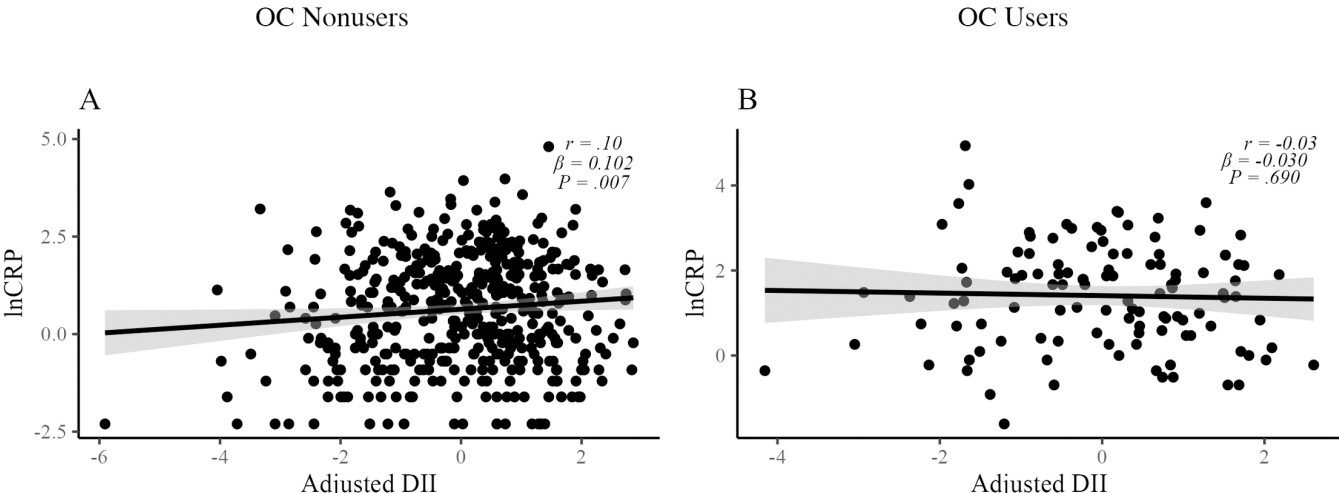

**Fig 4. Plots visualizing the independent relationships between DII and lnCRP after adjusting for smoking status, age, PA level, and BMI among OC nonusers (A) and OC users (B).** PA, physical activity; OC, oral contraceptive; BMI, body mass index; lnCRP, natural logarithm of C-reactive protein; DII, dietary inflammatory index.

values ($F[1, 490] = 192.64$, $p < .0001$). A plot visualizing the independent relationship between BMI and lnCRP after adjusting for smoking status, age, and PA level is presented in Fig 3D.

### Hypothesis 4

Full nutrition data were available for a subset of 578 participants. After adjusting for smoking status and age, PA level ($F[2, 458] = 3.92$, $p = .021$) was significantly predictive of lnCRP among OC nonusers ($n = 463$). When BMI was added to the model, it was also significantly predictive of lnCRP values ($F[1, 457] = 276.50$, $p < .0001$). After adjusting for all prior variables, DII was weakly (semipartial $r = .10$) but significantly predictive of lnCRP when added to the model ($F[1, 456] = 7.36$, $p = .007$). A plot visualizing the independent relationship between DII and lnCRP among OC nonusers, after adjusting for smoking status, age, PA level, and BMI, is presented in Fig 4A. After adjusting for smoking status and age, PA level was not significantly predictive of lnCRP values ($F[2, 110] = 0.34$, $p = .709$) among OC users ($n = 115$). When BMI was added to the model, BMI was significantly predictive of lnCRP values ($F[1, 109] = 42.40$, $p < .0001$). After adjusting for all prior variables, DII was not significantly predictive of lnCRP when added to the model ($F[1, 108] = 0.16$, $p = .690$). A plot visualizing the independent relationship between DII and lnCRP among OC users, after adjusting for smoking status, age, PA level, and BMI, is presented in Fig 4B.

### Sensitivity analysis

Menopause is typically reached by the age of 55, but estimates suggest that approximately 8% of individuals reach menopause after 55 [28], and credible case studies have documented premenopausal patients at 65 years of age [29]. The oldest participant in the present study was 61 years of age, and only 7 participants (0.3% of the sample) were above the age of 55. Sensitivity analyses were conducted to ensure that the inclusion of participants experiencing late menopause did not meaningfully alter study results. Replicating the statistical analysis after removing all participants above the age of 55 did not alter the outcomes of any statistical tests.

## Discussion

We hypothesized that 1) OC use and BMI would be positively associated with CRP levels in the full sample; 2) correlations between CRP and other indices of inflammation would be decoupled among OC users; 3) PA level would only be associated with CRP levels among OC nonusers; and 4) DII would only be associated with CRP levels among OC nonusers. Our findings lend support to hypotheses 1, 3, and 4, but contradict hypothesis 2. In line with previous research, CRP levels were positively associated with OC use and adiposity [30] but negatively associated with PA levels [12]. Contrary to our hypothesis, elevated CRP levels among OC users were correlated with alternative indices of inflammation and did not appear to be decoupled from systemic inflammation. PA level, BMI, and DII values were significantly predictive of circulating CRP values among OC nonusers in this sample. Body mass index was significantly predictive of circulating CRP values among OC users in this sample, but PA level and DII were not. Collectively, these results suggest that OC-induced CRP elevations do appear to reflect systemic inflammation but are not meaningfully attenuated by PA or anti-inflammatory diet patterns.

Previous studies focusing on bivariate relationships have provided strong evidence that CRP levels are positively associated with adiposity [30] and OC use [16], and inversely associated with PA level [12]. Fedewa et al. [13] hypothesized that PA would be associated with lower CRP levels in OC nonusers, but not in OC users. Their analysis of cross-sectional data from 247 female college students revealed a statistically significant three-way interaction partially supporting their hypothesis. Oral contraceptive use was associated with higher mean CRP levels across all categories of adiposity and PA level, and PA was only associated with substantial CRP reductions among OC nonusers with higher body-fat percentages. The present study did not find a statistically significant three-way interaction between PA level, OC use, and adiposity. Differing results may be related to the use of different outcomes variables for PA level (objectively measured step count versus self-reported activity level) and adiposity (body-fat percentage versus BMI), the use of different laboratory methods for CRP quantification (immunoturbidimetry versus nephelometry), or sampling from different populations (college students versus the general population). Nonetheless, the results of the present study are compatible with key findings reported by Fedewa et al. [13]. Both studies indicate that OC use is associated with marked CRP elevations, which do not appear to be meaningfully attenuated by increased PA. These observations are further supported by the findings of Cauci et al. [31], who reported a modest inverse association between exercise (hours per week) and CRP levels in Italian sportswomen who were OC nonusers. In contrast, exercise was not significantly associated with CRP levels among the OC users in this study.

Previous studies have reported surprisingly high CRP levels among lean, highly active athletes using OCs, with large CRP differences between athletes who use OCs in comparison to athletes who do not [15,31,32]. These studies have recruited samples of female athletes who compete at the national level [32], the international level [15], and a wide range of competitive levels [31]. Given that female endurance and team sport athletes tend to have low body-fat and engage in behaviors associated with low chronic inflammation levels, such as high levels of PA and consumption of a health-promoting diet [33], it has been hypothesized that these CRP elevations may not reflect systemic inflammation. Van Rooijen et al. [16] previously reported significant CRP elevations without concomitant increases in interleukin-6 or tumor necrosis factor α. As a result, the researchers concluded that CRP elevations may be driven by hepatic metabolism of orally ingested estrogen rather than systemic inflammation. This hypothesis is indirectly supported by research indicating that CRP elevations are specifically linked to the estrogen component of combined (estrogen-progestin) oral hormone formulations [18,19],

and that CRP elevations are attenuated or mitigated when provided via vaginal [20] or transdermal [21] routes of administration.

The present findings challenge the hypothesis that CRP elevations are decoupled from systemic inflammation among OC users. Correlations between CRP and other indices of systemic inflammation were weak in magnitude, but these relationships were significantly stronger among OC users in comparison to nonusers. While van Rooijen and colleagues previously reported that OC-induced CRP elevations were not associated with other biomarkers associated with systemic inflammation [16], contradictory studies have reported that OC-induced CRP elevations are correlated with blood hydroperoxides [22] and interleukin-6 [20]. Absolute risks of severe adverse events are low for OCs, but the primary adverse events associated with OC use are cardiovascular in nature, with OC users experiencing higher rates of venous thromboembolism, myocardial infarction, and stroke [34]. Notably, chronic inflammation is believed to play a causative role in the development or progression of all three of these cardiovascular complications [35].

In a 15-year study with over 1.6 million participants, Lidegaard et al. observed increased risk of thrombotic stroke and myocardial infarction among hormonal contraceptive users [34]. Risk elevations were positively associated with ethinyl estradiol dose, but differences among progestin types were minimal. Effect estimates for transdermal patches and vaginal rings were imprecise due to relatively low prevalence of use, but these contraceptive methods were associated with relative risk values (with 95% confidence intervals) of 3.15 (0.79, 12.60) and 2.49 (1.41, 4.41) for thrombotic stroke. Taken together, short-term biomarker data and long-term cardiovascular outcome data provide insufficient evidence to conclude that elevated CRP levels are decoupled from systemic inflammation levels among OC users, or that increases in CRP or systemic inflammation associated with exogenous estrogen use are exclusively observed with oral administration.

Current medical guidelines encourage risk stratification prior to prescription of OCs containing estrogen, which are contraindicated for individuals with multiple cardiovascular risk factors associated with chronic inflammation such as older age, diabetes, hypertension, and smoking [7]. The present findings support recommendations to conduct risk stratification before prescribing OCs containing estrogen, but also highlight the need for additional research to determine the clinical implications of OC-induced CRP elevations among otherwise healthy individuals. In the present study, median CRP values were 2.2 mg/L higher among OC users, and mean CRP values were 3.0 mg/L higher. A difference of this magnitude may be considered clinically meaningful. The median OC user in this sample would be considered "high risk," whereas the median nonuser would be considered "moderate risk" based on validated CRP cutoffs [5]. As presented in Table 1, OC users had a higher proportion of individuals with high cardiovascular risk (60.1% versus 38.5%) and a lower proportion of individuals with low cardiovascular risk (15.9% versus 33.4%) in comparison to nonusers. Even within risk categories, modest increases in CRP appear to be predictive of future cardiovascular event risk, with incremental risk increases associated with 1-unit CRP elevations from <0.50 to 5.0 mg/L [5]. Cardiovascular disease is the leading cause of death among U.S. women, with nearly 45% of American women over the age of 19 living with some form of cardiovascular disease [36]. In light of previous research associating OC use with increased risk of venous thrombosis, arterial thrombosis, myocardial infarction, and stroke [37], the clinical ramifications of OC-induced CRP elevations warrant further study.

Effective risk stratification reduces the likelihood of adverse cardiovascular effects among individuals with preexisting cardiometabolic risk factors, but lifestyle interventions that effectively attenuate OC-induced inflammation may be of clinical interest to low- or moderate-risk OC users. The present analysis did not identify any statistically significant interactions indicating that PA or diet pattern differentially impact CRP levels

among OC users or nonusers. The inverse association between PA level and CRP and the positive association between DII and CRP were statistically significant among OC non-users and nonsignificant among OC users, but these differences in nominal significance should not be interpreted as a statistically significant divergence between groups [38]. The observed associations among PA level, DII, and CRP were modest in magnitude, and the effect sizes associated with high PA and an anti-inflammatory diet pattern were not substantial enough to meaningfully attenuate the large CRP elevations observed among OC users. Future research should seek to identify effective interventions to attenuate OC-induced inflammation among healthy individuals.

The present study utilized a large, heterogeneous sample to address key gaps in the OC literature, but it must be interpreted with its limitations in mind. The study was designed to examine cross-sectional associations, so causal inferences should not be made. We excluded OC nonusers who did not experience a menstrual cycle within 2 months of taking the NHANES surveys, but it's possible that a small number of participants with menstrual cycle disorders were included in the sample. These surveys do not identify the current menstrual cycle phase at the time of each participant's NHANES assessment, but this unlikely to intro-duce a substantive or systematic bias to our analysis because CRP levels remain relatively stable across the menstrual cycle [24]. Surveys provided information about current OC use but did not provide details regarding duration of use or the exact type, formulation, and generation of OC used by each study participant. Nonetheless, these factors are unlikely to have a meaningful impact on our analysis. Crossover trials indicate that CRP elevations occur quickly during OC use and return to baseline quickly after cessation [16]. CRP responses are significantly attenuated by progestin-only formulations in comparison to combined formu-lations [19], but less than 1% of US women use progestin-only pills [39]. CRP responses do not appear to meaningfully differ when comparing among second, third, or fourth generation OCs [22], among active versus inactive pill phases [24], or among monophasic versus multi-phasic formulations [40].

Another limitation is that PA data were obtained from a self-reported survey rather than objective measurements, but this PA questionnaire was previously validated against accel-erometry data (total counts per day) in the 2003-2004 and 2005-2006 NHANES cycles [12]. It's also important to recognize that the sequential approach to model building in the pres-ent study was guided by the *a priori* hypotheses, but results may be sensitive to the order of variable entry due to the unbalanced design and correlations among predictor variables. Most importantly, the observed findings do not constitute adequate evidence to broadly discourage the use of OCs among athletes or the general population. Oral contraceptive pills are a highly effective method of contraception, may reduce menstrual symptoms, and do not appear to substantially impair athletic performance [41] or adaptations to exercise training [42]. While OCs are currently contraindicated for patients with high cardiovascular risk profiles, addi-tional research is needed to determine the clinical impact of chronic CRP elevations among otherwise healthy OC users.

In conclusion, OC use and BMI are associated with substantial elevations in CRP, and these elevations in CRP are associated with multiple alternative indices of systemic inflammation. PA is associated with a modest reduction in CRP levels, particularly among OC nonusers. Among OC users, neither PA nor anti-inflammatory eating patterns appear to meaningfully mitigate the CRP elevations associated with OC use. While medical practitioners are currently advised to conduct a comprehensive cardiovascular risk stratification prior to prescribing OCs, future research should aim to elucidate the clinical ramifications of OC-induced CRP elevations in otherwise healthy individuals. In addition, randomized controlled trials should be utilized to directly compare the effects of different hormonal contraceptive formulations

and routes of administration on inflammation biomarkers and to evaluate the efficacy of targeted interventions to attenuate systemic inflammation among OC users.

## Acknowledgements

We are grateful to the NHANES study participants, as well as the individuals and organizations responsible for collecting, organizing, and maintaining these valuable datasets.

## Author contributions

**Conceptualization:** Eric Trexler, Herman Pontzer.

**Formal analysis:** Eric Trexler.

**Methodology:** David E. Eagle.

**Supervision:** David E. Eagle, Herman Pontzer.

**Writing – original draft:** Eric Trexler.

**Writing – review & editing:** David E. Eagle, Herman Pontzer.

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
