## [Decision Letter · Decision Letter 0]

16 Jul 2024

PONE-D-24-18364Physical activity and diet pattern do not mitigate C-reactive protein increases associated with oral contraceptive usePLOS ONE

Dear Dr. Trexler,

Thank you for submitting your manuscript to PLOS ONE. After careful consideration, we feel that it has merit but does not fully meet PLOS ONE’s publication criteria as it currently stands. Therefore, we invite you to submit a revised version of the manuscript that addresses the points raised during the review process.

We look forward to receiving your revised manuscript.

Kind regards,

Samiullah Khan, Ph. D

Academic Editor

PLOS ONE

Additional Editor Comments :

Dear author,

Whole manuscript should be revised keeping in view all the comments of both reviewers.

Reviewers' comments:

Reviewer's Responses to Questions

**Comments to the Author**

1. Is the manuscript technically sound, and do the data support the conclusions?

Reviewer #1: Yes

Reviewer #2: Yes

2. Has the statistical analysis been performed appropriately and rigorously? 

Reviewer #1: Yes

Reviewer #2: Yes

3. Have the authors made all data underlying the findings in their manuscript fully available?

Reviewer #1: Yes

Reviewer #2: Yes

4. Is the manuscript presented in an intelligible fashion and written in standard English?

Reviewer #1: Yes

Reviewer #2: Yes

5. Review Comments to the Author

Reviewer #1: Thank you for this outstanding piece of work. The analysis and insights on this paper provide some key details on inflammation status in OCP users. The main finding of CRP values in OCP users not being decoupled from other markers of inflammation will be of interest to many in this field. I believe that this article will likely serve as the foundation for future research that seeks to understand the effects of this systemic inflammation on these women's health.

I have some minor comments that I would like the authors to address prior to the paper's acceptance.

1) Line 87-89: I felt that the sentence about possible mechanisms that could increase CRP levels in OCP users could do with some work. In this sentence 'but' could possibly be replaced with 'of which'.

2) Line 276: should 'oral contraceptives' be presented as OC as per the rest of the article?

Reviewer #2: This study investigated the influence of body mass index, self-reported physical activity level, dietary inflammatory index, and oral contractive use on C-reactive protein levels. It is a valuable study which provides further insight into the role of OC use in modulating CRP levels. The large sample size is a strength, however the reliance on retrospective and self-report data with limited detail to categorize participants is a limitation. Suggested changes below.

Introduction

-Very nice introduction! CRP is clearly introduced and its significance as an inflammatory biomarker explained. The prior research investigating oral contraceptive use on CRP concentrations has been summarized, and the ‘gaps’ and remaining questions have been clearly elucidated.

-However, it is stated that it is unclear “if these elevations [in CRP] can be mitigated or attenuated by physical activity, low BMI, or anti-inflammatory diet patterns”, yet you have discussed a study (Larsen et al 2020) showing significantly elevated CRP levels even in Olympic athletes with a low BMI (19.5), which seems to suggest that a low-normal BMI and high levels of PA will not attenuate CRP levels in OC users – this finding (also observed in other studies) needs further consideration/discussion in the Introduction to add clarity surrounding the aims and hypotheses of this research

Methods

-Line 128: were all OC users included? What about those on progestin-only pills? Was length of OC use captured?

-Line 157: it is stated that “first generation oral contraceptives are rarely used in the present day” however data was captured ~20 years ago. Implications of this should be discussed as a limitation

-Line 161: having a period every 2 months is not considered within the range of a regular MC. Thus, it is likely that subjects with MC dysfunction (and thus, a different hormonal profile to those with a regular MC) were captured in the study cohort. This also needs to be considered as a limitation

-Depending on responses to questions above, many different hormonal profiles could have been captured in the study i.e., those with regular MC, MC disorders, combined OC users, progestin only OC users… clarification and more detail is required around participant inclusion/exclusion criteria and the limitations thereof

-Were CRP samples collected at a specific phase of the MC or OC use (active or inactive pills)?

The Results are clearly presented – well done

Discussion:

-Why are only 3 of the 5 hypothesis presented in Para 1 of the Discussion? The numbers don’t align to those previously used either (1-3 vs 1-5). This is confusing – please re-write to align discussion with other manuscript sections when discussion hypotheses 1-5

-Otherwise, an excellent Discussion that explores the current findings with consideration of previous research without overstating the data. Limitations and considerations for future research are presented, limitations could be expanded as per suggestions above.

6. PLOS authors have the option to publish the peer review history of their article (what does this mean? ). If published, this will include your full peer review and any attached files.

**Do you want your identity to be public for this peer review?** For information about this choice, including consent withdrawal, please see our Privacy Policy .

Reviewer #1: **Yes: ** Claire Badenhorst

Reviewer #2: No

---

## [Author Response · Author response to Decision Letter 1]

29 Jul 2024

We would like to thank the editor and both reviewers for their insight and constructive feedback. We have addressed all reviewer comments with revisions that have strengthened the manuscript. Please find responses to all reviewer comments below.

Reviewer #1

1) Line 87-89: I felt that the sentence about possible mechanisms that could increase CRP levels in OCP users could do with some work. In this sentence 'but' could possibly be replaced with 'of which'.

We agree that this sentence lacked clarity in its original form. Thank you for bringing this to our attention. The statement now reads:

“There are several mechanisms by which OCs could increase CRP [16], which may include both systemic inflammation and localized effects on hepatic production of CRP.”

2) Line 276: should 'oral contraceptives' be presented as OC as per the rest of the article?

You are correct. We have updated the manuscript to ensure that “oral contraceptive” is abbreviated to “OC” in all instances following its first use (except at the beginning of new sentences, per common grammatical norms).

Reviewer #2

1) It is stated that it is unclear “if these elevations [in CRP] can be mitigated or attenuated by physical activity, low BMI, or anti-inflammatory diet patterns”, yet you have discussed a study (Larsen et al 2020) showing significantly elevated CRP levels even in Olympic athletes with a low BMI (19.5), which seems to suggest that a low-normal BMI and high levels of PA will not attenuate CRP levels in OC users – this finding (also observed in other studies) needs further consideration/discussion in the Introduction to add clarity surrounding the aims and hypotheses of this research

Thank you for highlighting this lack of clarity. The introduction has been revised as follows:

“Even among world-class endurance and team sport athletes, CRP levels are significantly elevated in OC users [15]. It is possible that acute stressors of intense training could contribute to these observed CRP elevations, but direct comparisons to OC nonusers engaged in similar training cast doubt on this potential explanation [15]. Preliminary studies appear to suggest that OC users are unable to fully mitigate elevations in CRP by maintaining low adiposity and high PA levels, but these observations are based on very few studies with relatively small sample sizes and homogenous populations, such as university students or athletic teams.”

“There are currently several unresolved questions pertaining to the clinical implications of CRP elevations induced by OC use. Most notably, it is unclear if OC-induced elevations in CRP reflect a systemic inflammatory response and if these elevations can be mitigated or attenuated by physical activity, low BMI, or anti-inflammatory diet patterns in the general population.”

2) Line 128: were all OC users included? What about those on progestin-only pills? Was length of OC use captured?

Thank you for prompting us to clarify these points. The manuscript now contains the following statements:

“The NHANES data do not distinguish between combined OCs (with synthetic estrogen and progestin) and progestin-only pills, so the present analysis includes all OC users. However, the inclusion of progestin-only pill users is unlikely to substantially impact the present analysis because less than 1% of US women use progestin-only pills [22].” (in methods section)

“The NHANES data do not distinguish between combined OCs and progestin-only pills, although this impact is likely negligible as less than 1% of US women use progestin-only pills [22].” (in limitations section)

“Survey questions about OC use specifically refer to current use, so duration of OC use is unknown for survey respondents.” (in methods section)

“Surveys provided information about current OC use but did not provide details regarding history and duration of use.” (in limitations section)

3) Line 157: it is stated that “first generation oral contraceptives are rarely used in the present day” however data was captured ~20 years ago. Implications of this should be discussed as a limitation

Thank you for highlighting this consideration. The severe side effects associated with first-generation contraceptives were identified in the 1960s and 1970s, which led to rapid adjustments to estrogen doses and the introduction of second- and third-generation progestins. As a result, first-generation contraceptives were largely phased out of clinical practice by the time these NHANES data were collected in 1999-2006. The manuscript now contains the following statement:

“These NHANES questionnaires provide incomplete information regarding the generation of contraceptive used by each study participant. However, this is unlikely to impact the present study’s findings because first generation OCs were largely phased out of clinical practice by the 1990s and CRP elevations appear to be similar among second, third, and fourth generation OCs [20].”

4) Line 161: having a period every 2 months is not considered within the range of a regular MC. Thus, it is likely that subjects with MC dysfunction (and thus, a different hormonal profile to those with a regular MC) were captured in the study cohort. This also needs to be considered as a limitation

Our decision to use this criterion was largely driven by the specific survey item used in NHANES. The survey asks participants “When did you have your last period?” The first two options (reflecting the two most recent answer options) are “Having it now” and “Less than 2 months ago.” This likely reflects clinically pertinent observations that 1) many females may have “normal” menstrual cycles up to 35 days in length, and 2) missing an occasional menstrual cycle does not meet typical criteria for amenorrhea or oligomenorrhea. Nonetheless, this is an important consideration to highlight in our paper. The manuscript now contains the following statements:

“Menstrual status was assessed by item RHQ051 (“When did you have your last period?”). For the present analysis, participants were considered “regularly menstruating” if they selected “having it now” or “less than 2 months ago” for this survey item, as answers beyond 2 months are more likely to reflect clinically relevant menstrual cycle disorders.” (methods section)

“Similarly, we excluded OC nonusers who did not experience a menstrual cycle within 2 months of taking the NHANES survey, but it’s possible that a small number of participants with menstrual cycle disorders were included in the sample.” (limitations section)

5) Depending on responses to questions above, many different hormonal profiles could have been captured in the study i.e., those with regular MC, MC disorders, combined OC users, progestin only OC users… clarification and more detail is required around participant inclusion/exclusion criteria and the limitations thereof

We agree that these considerations are important to highlight in the manuscript. The limitations section now indicates:

“Surveys provided information about current OC use but did not provide details regarding history and duration of use. Similarly, we excluded OC nonusers who did not experience a menstrual cycle within 2 months of taking the NHANES survey, but it’s possible that a small number of participants with menstrual cycle disorders were included in the sample. The NHANES data do not distinguish between combined OCs and progestin-only pills, although this impact is likely negligible as less than 1% of US women use progestin-only pills [22]. These surveys do not identify the current menstrual cycle phase or distinguish between the active or inactive phase of OC pills at the time of each participant’s NHANES assessment, although is unlikely to introduce a substantive or systematic bias to our analysis [23]. The NHANES data also fail to specify the specific generation of OC being used by each participant, but CRP elevations are consistently observed across the generations that are currently used in clinical practice [20].”

6) Were CRP samples collected at a specific phase of the MC or OC use (active or inactive pills)?

Due to the large and cross-sectional nature of the NHANES study, it was not feasible to standardize participants’ MC phase or OC (active versus inactive) phase at the time of assessment. In addition, NHANES surveys do not capture this information. Fortunately this is unlikely to introduce systematic bias to our analysis, and previous literature indicates that CRP fluctuations across the menstrual cycle are small in magnitude and that OC-induced CRP elevations persist across all menstrual cycle phases and OC pill phases. We have added the following statements to our manuscript to reflect this information:

“Survey items related to OC use and menstrual status do not identify the current menstrual cycle phase or distinguish between the active or inactive phase of OC pills at the time of each participant’s NHANES assessment. However, this is unlikely to introduce systematic bias to our analysis, and previous literature indicates that CRP fluctuations across the menstrual cycle are small in magnitude and that OC-induced CRP elevations persist across all menstrual cycle phases and OC pill phases [23].” (in methods section)

“These surveys do not identify the current menstrual cycle phase or distinguish between the active or inactive phase of OC pills at the time of each participant’s NHANES assessment, although is unlikely to introduce a substantive or systematic bias to our analysis [23].” (in limitations section)

Discussion

7) Why are only 3 of the 5 hypothesis presented in Para 1 of the Discussion? The numbers don’t align to those previously used either (1-3 vs 1-5). This is confusing – please re-write to align discussion with other manuscript sections when discussion hypotheses 1-5

Thank you for this suggestion. We agree that the clarity of the paper would be improved by refining the numbering of hypotheses and maintaining consistency when discussing them in order. To eliminate redundancy from hypotheses with considerable overlap, we have condensed the list of hypotheses to 4, and these same 4 numbered hypotheses are discussed in order in the introduction section, the statistical analysis section, the results section, and the first paragraph of the discussion section. The manuscript has been revised accordingly.

The introduction section now reads:

“We hypothesized that 1) OC use and BMI would be positively associated with CRP levels in the full sample; 2) correlations between CRP and other indices of inflammation would be decoupled among OC users (reflecting OC-induced CRP elevations in the absence of a systemic inflammatory response); 3) PA level would only be associated with CRP levels among OC nonusers; and 4) DII would only be associated with CRP levels among OC nonusers. These hypotheses collectively reflect the perspective that OC-induced elevations in CRP are not indicative of systemic inflammation, and are therefore unresponsive to the anti-inflammatory effects of PA and low DII.”

The discussion section now reads:

“We hypothesized that 1) OC use and BMI would be positively associated with CRP levels in the full sample; 2) correlations between CRP and other indices of inflammation would be decoupled among OC users; 3) PA level would only be associated with CRP levels among OC nonusers; and 4) DII would only be associated with CRP levels among OC nonusers. Our findings lend support to hypotheses 1, 3, and 4, but contradict hypothesis 2. In line with previous research, CRP levels were positively associated with OC use and adiposity [29] but negatively associated with PA levels [12]. Contrary to our hypothesis, elevated CRP levels among OC users were correlated with alternative indices of inflammation and did not appear to be decoupled from systemic inflammation. Physical activity level, BMI, and DII values were significantly predictive of circulating CRP values among OC nonusers in this sample. Body mass index was significantly predictive of circulating CRP values among OC users in this sample, but PA level and DII were not. Collectively, these results suggest that OC-induced CRP elevations do appear to reflect systemic inflammation but are not meaningfully attenuated by physical activity or anti-inflammatory diet patterns.”

---

## [Decision Letter · Decision Letter 1]

28 Aug 2024

PONE-D-24-18364R1Physical activity and diet pattern do not mitigate C-reactive protein increases associated with oral contraceptive usePLOS ONE

Dear Dr. Trexler

Thank you for submitting your manuscript to PLOS ONE. After careful consideration, we feel that it has merit but does not fully meet PLOS ONE’s publication criteria as it currently stands. Therefore, we invite you to submit a revised version of the manuscript that addresses the points raised during the review process.

We look forward to receiving your revised manuscript.

Kind regards,

Samiullah Khan, Ph. D

Academic Editor

PLOS ONE

Journal Requirements:

Additional Editor Comments:

Dear Editor,

The changes have substantially improved the presentation of the work. The reviewer #1 had some additional comments that I would like the authors to consider and address prior to the paper being accepted for publication.

Reviewers' comments:

Reviewer's Responses to Questions

**Comments to the Author**

1. If the authors have adequately addressed your comments raised in a previous round of review and you feel that this manuscript is now acceptable for publication, you may indicate that here to bypass the “Comments to the Author” section, enter your conflict of interest statement in the “Confidential to Editor” section, and submit your "Accept" recommendation.

Reviewer #1: All comments have been addressed

Reviewer #2: All comments have been addressed

2. Is the manuscript technically sound, and do the data support the conclusions?

Reviewer #1: Yes

Reviewer #2: Yes

3. Has the statistical analysis been performed appropriately and rigorously? 

Reviewer #1: Yes

Reviewer #2: Yes

4. Have the authors made all data underlying the findings in their manuscript fully available?

Reviewer #1: Yes

Reviewer #2: Yes

5. Is the manuscript presented in an intelligible fashion and written in standard English?

Reviewer #1: Yes

Reviewer #2: Yes

6. Review Comments to the Author

Reviewer #1: Thank you to the authors for their edits to their paper. The changes have substantially improved the presentation of the work. I do have some additional comments that I would like the authors to consider and address prior to the paper being accepted for publication. I hope that the authors find these helpful.

Comments:

1) It is noted that the authors define the abbreviations in the abstract of the paper. However, in the introduction, the authors again define the abbreviations CPR and OC but do not redefine PA, BMI or DII. This may need to be reviewed.

2) Should line 86 state ‘that physical and body composition are insufficient for mitigating systemic inflammation’? The addition of body composition would align with the information provided in the previous paragraph.

3) Line 90: can the authors check and clarify if it is ‘both’ systemic inflammation and localised effects or is it ‘either’?

4) When referring to synthetic estrogen the authors may consider using ethylestradiol vs estradiol (lines 91 ,94). This change in terminology may also be considered and edited throughout the paper.

5) Line 113: I am not sure why the authors have selected their hypotheses based on OC- induced elevations not being associated with systemic inflammation. The information provided by the authors would suggest that the previous evidence not in favour of systemic inflammation is a) done in small homogenous samples (line 83) or is equivocal (line 96). The authors may need to consider a revision of their statement that justifies why hypothesis 2 was presented.

6) Line 155: should this statement include type and duration? While the authors have noted that less than 1% of US women may use progestin-only pills, there are different types (mono-,bi-, and tri-phasic OCPs) that women in the study may be using.

7) Line 165: can the authors confirm if this statement of menstrual cycle status was in the later 4 years of the NHANES data? The previous section outlines differences between the data collection periods and the information here would be very helpful to the reader.

8) Why is information on HRT presented between lines 170-173? Would it not be better suited to the paragraph above which specifically focuses on OCs data collection from the NHANES survey?

9) Line 177: will need to be edited and have ‘menstrual cycle phases removed’. If an individual is using an OC then they do not have a menstrual cycle they will only have active and pill-free phases.

10) Physical activity’s abbreviation was defined in the abstract, however, there is inconsistent use of this abbreviation throughout the article. The authors may need to review this. Lines where the abbreviation has not been used have been identified: 86, 387, 391, 401, 411, and 490.

Reviewer #2: The authors have made all suggested comments and the manuscript has been significantly improved as a result, well done. This paper will hopefully prompt further research into the implications of elevated CRP concentrations amongst OC users - an important topic given the high prevalence of OC use amongst women.

7. PLOS authors have the option to publish the peer review history of their article (what does this mean? ). If published, this will include your full peer review and any attached files.

**Do you want your identity to be public for this peer review?** For information about this choice, including consent withdrawal, please see our Privacy Policy .

Reviewer #1: No

Reviewer #2: No

---

## [Author Response · Author response to Decision Letter 2]

24 Sep 2024

We would like to thank the editor and reviewers for their insight and constructive feedback. We have addressed all reviewer comments with revisions that have strengthened the manuscript. Please find responses to all reviewer comments below.

Reviewer #1

Comment: Thank you to the authors for their edits to their paper. The changes have substantially improved the presentation of the work. I do have some additional comments that I would like the authors to consider and address prior to the paper being accepted for publication. I hope that the authors find these helpful.

Response: Thank you for your time, effort, and careful attention to detail throughout this review process. We believe this additional round of revisions has improved the manuscript even further.

1) Comment: It is noted that the authors define the abbreviations in the abstract of the paper. However, in the introduction, the authors again define the abbreviations CPR and OC but do not redefine PA, BMI or DII. This may need to be reviewed.

Response: Thank you for highlighting this discrepancy. We believe it is helpful to define acronyms at first use in both the abstract and the full text, so in the interest of consistency we have updated the manuscript to define PA, BMI, and DII in the introduction section as well.

2) Comment: Should line 86 state ‘that physical and body composition are insufficient for mitigating systemic inflammation’? The addition of body composition would align with the information provided in the previous paragraph.

Response: Thank you for this suggestion. The sentence has been updated as follows:

“Given the well-established link between CRP and systemic inflammation, the persistence of OC-induced CRP elevations in lean and highly active athletes may suggest that PA and low adiposity are insufficient for mitigating systemic inflammation induced by OCs.”

3) Comment: Line 90: can the authors check and clarify if it is ‘both’ systemic inflammation and localised effects or is it ‘either’?

Response: Thank you for identifying this lack of clarity. The intended meaning of the statement is best reflected by the use of the word “or,” so the sentence has been updated as follows:

“There are several mechanisms by which OCs could increase CRP [17], which may include systemic inflammation or localized effects on hepatic production of CRP.”

4) Comment: When referring to synthetic estrogen the authors may consider using ethylestradiol vs estradiol (lines 91 ,94). This change in terminology may also be considered and edited throughout the paper.

Response: Thank you for prompting more careful consideration of this terminology. We believe it’s important for this terminology to provide as much specificity as possible without sacrificing accuracy. We have carefully revised the manuscript to maximize the specificity of our terminology regarding estrogens. When referring to the results of any intervention using a specific form of estrogen, we use the specific name of the form (i.e., ethinyl estradiol, estetrol, estradiol valerate, mestranol, etc.) used in the study. When referring to generalized statements that apply to all forms of currently used estrogens, or scenarios in which the exact form cannot be definitively ascertained, we use the generic term “estrogen,” with further specification (when necessary) as to whether we are referring to exogenous or endogenous estrogen.

5) Comment: Line 113: I am not sure why the authors have selected their hypotheses based on OC- induced elevations not being associated with systemic inflammation. The information provided by the authors would suggest that the previous evidence not in favour of systemic inflammation is a) done in small homogenous samples (line 83) or is equivocal (line 96). The authors may need to consider a revision of their statement that justifies why hypothesis 2 was presented.

Response: We agree that the foundation for hypothesis 2 requires additional explanation and justification. We have added the following text to clarify:

“Given the well-established link between CRP and systemic inflammation, the persistence of OC-induced CRP elevations in lean and highly active athletes may suggest that PA and low adiposity are insufficient for mitigating systemic inflammation induced by OCs. Alternatively, these observations of high CRP levels in athletes may suggest that OC-induced CRP elevations are not reflective of systemic inflammation. In support of this concept, a randomized crossover trial by Van Rooijen et al. [16] previously reported significant CRP elevations without concomitant increases in interleukin-6 or tumor necrosis factor alpha in response to a combined OC containing ethinyl estradiol, which led the researchers to conclude that the observed CRP elevations were not indicative of a systemic inflammatory response.”

6) Comment: Line 155: should this statement include type and duration? While the authors have noted that less than 1% of US women may use progestin-only pills, there are different types (mono-,bi-, and tri-phasic OCPs) that women in the study may be using.

Response: Thank you for this suggestion. The section has been revised to improve clarity as follows:

“These NHANES questionnaires also provide incomplete information regarding the exact type, formulation, and generation of OC used by each study participant. As a result, the present analysis is not able to distinguish between progestin-only versus combined formulations, monophasic versus multiphasic formulations, different forms or generations of exogenous estrogen and progestin, or different dosages of exogenous estrogen and progestin.”

We have also updated the limitations section to acknowledge this point as concisely as possible. It now reads:

“These surveys do not identify the current menstrual cycle phase at the time of each participant’s NHANES assessment, but this unlikely to introduce a substantive or systematic bias to our analysis because CRP levels remain relatively stable across the menstrual cycle [24]. Surveys provided information about current OC use but did not provide details regarding duration of use or the exact type, formulation, and generation of OC used by each study participant. Nonetheless, these factors are unlikely to have a meaningful impact on our analysis. Crossover trials indicate that CRP elevations occur quickly during OC use and return to baseline quickly after cessation [16]. CRP responses are significantly attenuated by progestin-only formulations in comparison to combined formulations [19], but less than 1% of US women use progestin-only pills [39]. CRP responses do not appear to meaningfully differ when comparing among second, third, or fourth generation OCs [22], among active versus inactive pill phases [24], or among monophasic versus multiphasic formulations [40].”

7) Comment: Line 165: can the authors confirm if this statement of menstrual cycle status was in the later 4 years of the NHANES data? The previous section outlines differences between the data collection periods and the information here would be very helpful to the reader.

Response: Thank you for prompting this clarification. The content of this survey item remained unchanged from 1999-2006, but the identification number of the survey item was changed from RHQ050 to RHQ051 in 2003. The texted has been updated to clarify this change in numbering:

“Menstrual status was assessed by a survey item that asked participants, ‘When did you have your last period?’ This survey item was identified as RHQ050 from 1999-2002 and as RHQ051 from 2003-2006.”

8) Comment: Why is information on HRT presented between lines 170-173? Would it not be better suited to the paragraph above which specifically focuses on OCs data collection from the NHANES survey?

Response: Thank you for this suggestion. We have moved this information about HRT to the prior paragraph as directed.

9) Comment: Line 177: will need to be edited and have ‘menstrual cycle phases removed’. If an individual is using an OC then they do not have a menstrual cycle they will only have active and pill-free phases.

Response: Thank you for bringing this edit to our attention. It has been implemented as directed.

10) Comment: Physical activity’s abbreviation was defined in the abstract, however, there is inconsistent use of this abbreviation throughout the article. The authors may need to review this. Lines where the abbreviation has not been used have been identified: 86, 387, 391, 401, 411, and 490.

Response: Thank you for identifying this oversight. We have checked and corrected the article to ensure that all abbreviations have been used consistently throughout the manuscript.

Reviewer #2

1) Comment: The authors have made all suggested comments and the manuscript has been significantly improved as a result, well done. This paper will hopefully prompt further research into the implications of elevated CRP concentrations amongst OC users - an important topic given the high prevalence of OC use amongst women.

Response: Thank you for your positive and constructive feedback throughout the review process.

---

## [Decision Letter · Decision Letter 2]

11 Feb 2025

Physical activity and diet pattern do not mitigate C-reactive protein increases associated with oral contraceptive use

PONE-D-24-18364R2

Dear Dr Eric Trexler,.,

We’re pleased to inform you that your manuscript has been judged scientifically suitable for publication and will be formally accepted for publication once it meets all outstanding technical requirements.

Kind regards,

Samiullah Khan, Ph. D

Academic Editor

PLOS ONE

Additional Editor Comments (optional):

Reviewers' comments:

Reviewer's Responses to Questions

**Comments to the Author**

1. If the authors have adequately addressed your comments raised in a previous round of review and you feel that this manuscript is now acceptable for publication, you may indicate that here to bypass the “Comments to the Author” section, enter your conflict of interest statement in the “Confidential to Editor” section, and submit your "Accept" recommendation.

Reviewer #1: All comments have been addressed

2. Is the manuscript technically sound, and do the data support the conclusions?

Reviewer #1: Yes

3. Has the statistical analysis been performed appropriately and rigorously? 

Reviewer #1: Yes

4. Have the authors made all data underlying the findings in their manuscript fully available?

Reviewer #1: Yes

5. Is the manuscript presented in an intelligible fashion and written in standard English?

Reviewer #1: Yes

6. Review Comments to the Author

Reviewer #1: Thank you for completing the additional edits to this paper. It is a great piece of work and I look forward to seeing how it is received.

One final edit that may have been missed is the use of CRP abbreviation on line 332.

7. PLOS authors have the option to publish the peer review history of their article (what does this mean? ). If published, this will include your full peer review and any attached files.

**Do you want your identity to be public for this peer review?** For information about this choice, including consent withdrawal, please see our Privacy Policy .

Reviewer #1: **Yes: ** Claire Badenhorst

---

## [Editor Report · Acceptance letter]

PONE-D-24-18364R2

PLOS ONE

Dear Dr. Trexler,

I'm pleased to inform you that your manuscript has been deemed suitable for publication in PLOS ONE. Congratulations! Your manuscript is now being handed over to our production team.

Kind regards,

on behalf of

Dr. Samiullah Khan

Academic Editor

PLOS ONE